# Potential Role of Laccases in the Relationship of the Maize Late Wilt Causal Agent, *Magnaporthiopsis maydis*, and Its Host

**DOI:** 10.3390/jof6020063

**Published:** 2020-05-17

**Authors:** Ofir Degani, Yuval Goldblat

**Affiliations:** 1Plant Sciences Department, MIGAL–Galilee Research Institute, Tarshish 2, Kiryat Shmona 11016, Israel; uvgold@gmail.com; 2Faculty of Sciences, Tel-Hai College, Upper Galilee, Tel-Hai 12210, Israel

**Keywords:** *Cephalosporium maydis*, corn, cotton, enzyme, fungus, *Harpophora maydis*, host–pathogen interactions, lacasse, roots pathogenicity assay, watermelon

## Abstract

Late wilt is a vascular disease of maize (*Zea mays* L.) caused by the soil-borne and seed-borne fungus *Magnaporthiopsis maydis.* The pathogen penetrates the roots of maize plants at the seedling stage, grows into the xylem vessels, and gradually spreads upwards. From the flowering stage to the kernel ripening, the fungal hyphae and secreted materials block the water supply in susceptible maize cultivars, leading to rapid dehydration and death. Laccase is an enzyme secreted by fungus for diverse purposes. The *M. maydis* laccase gene was identified in our laboratory, but under what conditions it is expressed and to what functions remain unknown. In the current study, we tested the influence of plant age and tissue source (roots or leaves) on *M. maydis* laccase secretion. The results show increasing laccase secretion as corn parts (as ground tissue) were added to the minimal medium (MM). Furthermore, roots stimulated laccase secretion more than leaves, and adult plants enhanced laccase secretion more than young plants. This implies the possibility that the richer lignin tissue of adult plants may cause increased secretion of the enzyme. In vitro pathogenicity assay proved the ability of *M. maydis* to develop inside detached roots of maize, barley, watermelon, and cotton but not peanut. Testing root powder from those plants in MM revealed a negative correlation between *M. maydis* growth (expressed as biomass) and laccase secretion. For example, while the addition of maize, barley, or cotton root powder led to increasing fungal dry weight, it also resulted in relatively lower laccase activity. Watermelon and peanut root powder led to opposite responses. These findings suggest a pivotal role of laccase in the ability of *M. maydis* to exploit and grow on different host tissues. The results encourage further examination and a deeper understanding of the laccase role in these interesting host–pathogen interactions.

## 1. Introduction

The maize late wilt disease causes severe damage to cornfields throughout Israel. The disease is characterized by rapid wilt of sweet and fodder maize, mainly from the tasseling stage phase until shortly before maturity [1]. The causal agent of the disease is the phytopathogenic fungus *Magnaporthiopsis maydis* (synonyms are *Harpophora maydis* and *Cephalosporium maydis*) [2]. The pathogen is a hemibiotroph, soil-borne [3], and seed-borne [4] fungus transmitted as spores, sclerotia, or hype on plant residues. *M. maydis* can survive in the soil (for long periods) or by developing inside an alternative host plant, such as *Lupinus termis* L. (lupine) [5], *Gossypium hirsutum* L. (cotton), *Citrullus lanatus* (watermelon), and *Setaria viridis* (green foxtail) [6,7].

Late wilt disease is considered the primary threat to maize fields in Egypt [8] and Israel [9] and a serious concern to other countries [10,11,12]. Attempts had been made in the past to control the disease using agricultural (balanced soil fertility and flood fallowing) [13,14], biological [15], physical (solar heating) [16], allelochemical [17], and chemical options [9,18,19], with different degrees of success. Nevertheless, today, the primary means of coping with the disease is the use of reduced sensitivity maize cultivars. Recently, for the first time, since the discovery of late wilt disease in Israel, an economical and efficient applicable solution was approved that can be used on a large scale to protect sensitive maize varieties in commercial fields [20].

The pathogen mode of infection is well detailed in the literature observations [4,12,21,22,23]. According to Sabet et al. [22], the infection takes place during the first three weeks of growth. After penetrating the roots, the fungus first appears in the xylem 21 days after seeding. The pathogen reaches the first stem internode above the ground after 35 days and the fourth internode after 49 days. At that stage (nearly 50 days from sowing), *M. maydis* DNA is detected in various parts of the host plant, with the highest concentration being in the roots [24]. When tassels first emerged (day 63) [22], the fungus was found throughout the length of the stalk, although there was less of it towards the top of the plant. At this plant age, the DNA levels peaked in the stems of a susceptible variety [24], and the first disease symptoms appeared shortly after that. The first symptom is moderately rapid plant wilting [23], progressing upwards. Leaves gradually lose their color and become dehydrated. At 78–84 days post sowing, the fungus was detected in the cob stalks and kernels. The abovementioned pathogenesis progression is described for a sensitive, sweet maize cultivar, whereas a similar infection mode occurs in a resistant cultivar but with about a two-week delay [24]. 

Laccases (benzenediol: oxygen oxidoreductase, EC 1.10.3.2) are typically produced enzymes in plants and fungi and in some bacteria. In plants, laccases participate in the radical-based mechanisms of lignin polymer formation. Fungi in general and phytopathogenic fungi in particular secrete this enzyme for a variety of purposes. Some functions are related to fungal growth and development, and others are related to interactions with the host plant [25]. Laccases are a group of oxidizing enzymes that catalyzes the reduction of molecular oxygen to water, bypassing the stage of hydrogen peroxide production [26], and they have a variety of potential applications in industry and biotechnology processes [27]. Laccases can oxidize a variety of organic (mainly aromatic) components as well as several inorganic components, including lignin.

Fungal laccases have many other roles, including defensive stress response and an offensive role to plants [28]. However, some of its biological functions are unknown. It was suggested that laccase’s major role in the fungus’ interactions with the host plant is the enzymatic decomposition of plant-produced toxic metabolites that are dangerous to fungus [29]. Indeed, laccase neutralizes the toxicity of these low molecular weight compounds in plants, making them nontoxic to the fungal hyphae. Different fungi were identified to contain closely regulated laccase genes, and phytopathogenic ascomycetes fungi were reported to produce multiple laccase isoforms (isoenzymes). For example, secretion of three laccase isoenzymes was found in *Gaeumannomyces graminis* [30]. 

Laccases are generally produced during the secondary metabolism of different fungi growing on a natural substrate or in submerged cultures [31]. Various cultivation parameters influence laccase production, such as carbon limitation, nitrogen source, and concentration, and the composition of microelements. In contrast, the use of excessive concentrations of glucose as a carbon source in the cultivation of laccase-producing fungal strains has an inhibitory effect on laccase production known as carbon catabolite repression [32].

The gene encoding laccase in *M. maydis* was previously identified and sequenced in our laboratory [33]. Still, it is unknown under what conditions the enzyme is produced and secreted and for what purpose (fungal development or the interactions with the host plant). Answering these questions is significant for our understanding of this emerging pathogen morphogenesis, its host relations, and its mode of operation during the infection process. The current work is the first report on *M. maydis* laccase secretion and production under the influence of a host tissue in submerged cultures. To this end, we used in vitro growth media manipulations and roots pathogenicity assay and measured variations in fungal growth weight, DNA levels, and extracellular laccase activity.

## 2. Materials and Methods

### 2.1. Fungal Isolates and Growth Conditions

One selected isolate of *M. maydis*, *Hm*2 (CBS 133,165), was used for this study. This isolate is currently deposited at the CBS-KNAW Fungal Biodiversity Center, Utrecht, The Netherlands. The morphological, microscopic, and molecular characteristics of the *Hm*2 isolate have all been well studied and have confirmed its identification [24,34]. All fungal colonies were grown on solid potato dextrose agar (PDA) medium (Difco, Detroit, MI, USA) at 28 ± 1 °C in the dark for 4–6 days. For submerged culturing, five 6-mm-diameter agar disks cut from the margins of PDA *M. maydis* colony (grown under the above conditions) were added to each 50 mL sterile minimal medium (MM) in a 250-mL Erlenmeyer flask. The MM was prepared according to Reference [35] and contained micro- and macronutrients without glucose or with 0.1% glucose (as will be indicated for each experiment) in double-distilled water (DDW). 

To these flasks, 750 mg of plant parts (roots or leaves) powder (prepared by grinding in liquid nitrogen) was added. Cultures were grown at 28 ± 1 °C in complete darkness on a rotary shaker at 150 revolutions per minute (RPM) for one week. At the end of the growth, the fungus and plant components were filtered from the substrate through double-layer Whatman 3-mm filter paper in a Büchner funnel, dried at 40 °C for 24 h to a nearly complete dryness and weighed.

### 2.2. Maize and Other Plant Species Seedling Growth

The maize cultivars chosen for the experiments were Jubilee cv. (sweet maize from SRS Snowy River seeds, Australia, supplied by Green 2000 Ltd., Bitan Aharon, Israel) and Prelude cv. (from SRS Snowy River seeds, Australia, supplied by Green 2000 Ltd., Bitan Aharon, Israel), as will be indicated for each figure. Both cultivars had previously been tested for susceptibility in the field and proved to be late wilt sensitive [1,9,20,24]. Other summer field crops tested in this work were *Gossypium hirsutum* L. (cotton, Pima cv.), *Citrullus lanatus* (watermelon, Malali cv.), *Hordeum vulgare* (barley, Noga cv.), and *Arachis hypogaea* (peanut or groundnut, Harari cv.). Cotton and watermelon were previously proven to be *M. maydis* hosts [6,7]. All the plants were grown for 30 days, in a growth chamber at a constant temperature of 27 ± 1 °C and relative humidity of 38% under a 16-h photoperiod illuminated by cool-white fluorescent tubes (Philips, Eindhoven, The Netherlands). The soil was commercial, non-sterilized peat mixed with 30% Perlite No. 4 (for aerating the ground). Irrigation was done by adding 100 ± 10 mL tap water every 48 h to the pots using a computerized irrigation system.

### 2.3. Laccase Activity Assay

The fungus was incubated in the test medium for various periods, and the resultant suspension was filtrated and tested with 2,2′-azinobis (3-ethylbenzothiazoline-6-sulfonic acid) (ABTS, Sigma-Aldrich Co., St. Louis, MO, USA). This substrate oxidized at high efficiency by laccase to produce a bluish-green color product measured in this work at 450 nm wavelength. Today, the majority of works described in the literature are done with 420 nm, which is considered the optimal wavelength for the ABTS substrate. Thus, although the current work was conducted near this optimum wavelength (at 450 nm), the measurements were still strong enough and differed significantly from each other (*p* < 0.05) to allow an accurate and detailed understanding of the host tissue effect on *M. maydis* extracellular laccase activity.

The growth fluid from each treatment was filtered through a 0.2-micron filter to separate the fungus from the substrate. From each repetition, 237.5 µl was moved to 96-well microplates (Greiner Bio-One GmbH, Frickenhausen, Germany). To each well, 12.5 µl of 50 mM ABTS (dissolved in 50 mM Buffer Citrate, pH 5) was added. Absorbance changes were measured by HP 8453 UV-visible Spectroscopy Systems (Hewlett Packard GmbH, Waldbronn, Germany) after incubation at 28 ± 1 °C in the dark for for 10 min (unless otherwise indicated). 

The enzymatic units (U) was defined as the amount of enzyme oxidizing 1 µmol ABTS in 1 min and was calculated by the following formula: *U*/*L* = (∆A × *Vt* × 10^6^)/(ε × *d* × *Vs*) [36,37], where ∆A = change in absorbance per min; Vt = total volume measured; 10^6^ = correction factor; Vs = volume of enzyme; (ε) of ABTS = extinction coefficient for the oxidation, which is 36,000 M-1 cm^−1^; and d = path length of the optical cell, which is 1 cm.

### 2.4. In Vitro Root Infection

Lateral, young, white, about 2-cm-long roots were removed from 30-day-old seedlings and washed under tap water to discharge the soil. Immediately afterward, they were sterilized using 70% ethanol solution and dried using a fume hood. Each root was moved to a sterile Petri dish covered on the inside bottom with Whatman 3-mm filter paper. Two milliliters of autoclaved DDW was used for wetting the filter paper. Culture PDA agar disk (6 mm in diameter) cut from the growing edge of a 4–6-day-old *M. maydis* colony (grown previously at 28 ± 1 °C in the dark) was placed at the cut end of each root. In the negative control roots, a sterile PDA agar disk was used instead. The Petri plates were maintained in the dark at 28 ± 1 °C. Each treatment was conducted in five independent repetitions, and the entire experiment was conducted twice. The lengths of root infection threads (a dark filament within the plants’ detached root) were identified, measured, and photographed three and six days after inoculation. For DNA extraction, one fragment was cut 0.5 cm from the infected (or uninfected in the control treatment) roots’ cut ends. DNA isolation and qPCR were conducted, as will be described below.

### 2.5. Molecular Diagnosis of Magnaporthiopsis Maydis

Each root fragment considered one replication. DNA was obtained using the Extract-N-amp plant PCR kit (Sigma-Aldrich Co., St. Louis, MO, USA) according to the manufacturer’s instructions. Each repeat was tested three times using qPCR to ensure consistency of the results. The qPCR method used was recently described in detail [6]. Briefly, the qPCR reactions were executed using the ABI PRISM^®^ 7900 HT Sequence Detection System (Applied Biosystems, Foster City, CA, USA) for 384-well plates. A reaction volume of 5 µL (in total) contained 2 µL of the DNA extract sample, 2.5 µL of the Universal SYBR^®^ Green Supermix iTaq™ (Bio-Rad Laboratories Ltd., Rishon Le Zion, Israel), and 0.25 µL of each of the forward and reverse primers (at a concentration of 10 µM of each primer per well). The qPCR plan was 1 min at 95 °C precycle activation stage; 40 cycles of denaturation (15 s at 95 °C), and then annealing and extension (30 s at 60 °C), followed by a melting curve. The target A200a *M. maydis*-specific DNA was evaluated against a reference “housekeeping” gene—the mitochondria-cytochrome c oxidase, *COXI* gene (sequences in Table 1).

### 2.6. Statistical Analyses

A completely randomized statistical design was used in all experiments. Data analysis and statistics were done using the JMP program, 15th edition, SAS Institute Inc., Cary, NC, USA. The one-way analysis of variance (ANOVA) followed by multiple comparisons post hoc of the Student’s *t*-test for each pair was used to evaluate the *M. maydis* infection outcome in the experiments. The *t*-test compared each treatment to the control (with a significance threshold of *p* = 0.05). Relatively high standard error values were expected in the detached root inoculation experiment qPCR results, whereas an objective difficulty in achieving a uniform infection exists.

## 3. Results

Two sets of experiments were conducted to reveal the host tissue rule in *M. maydis* laccase secretion. The first experimental set aimed at studying the effect of maize sprouts’ tissues at different ages on laccase secretion in *M. maydis*. The second goal was to answer the question of whether different hosts’ root tissues have a distinct influence on laccase secretion. 

In the first experiment, root powder from a 40-day-old plant in a minimal liquid medium was used as a nutrition source (Figure 1). Changes in laccase activity over time expressed as increasing product concentration (U/L) produced a linear curve. This ensured that the substrate used (ABTS) was at an excessive concentration and was not consumed during this time point. Also, the curve showed that the selected time point (10 min) was within the linear curve zone. Therefore, we conducted all the laccase assays at this time point. 

In leaf-supplemented media, the fresh fungal biomass was significantly (*p* < 0.05) higher compared to both aged root-containing media (Figure 2). However, this response was dependent on plant tissue age. The 43-day-old plants’ leaf-supplemented medium caused the maximum fungal growth, whereas in medium containing 23-day-old plants’ aboveground parts powder, a significant (*p* < 0.05) 3.4-fold reduction in fungal biomass was recorded. In the root-containing media, no age-dependent response was recorded, and both ages tested (23 and 43 days) led to lower and similar fungal wet weights.

According to Figure 1, an incubation time of 10 min was set in the laccase assay in submerged cultures, which enables the detection of differences in laccase secretion between leaf and root. Indeed, in a follow-up experiment, we investigated the effect of plant age on extracellular laccase secretion in *M. maydis* (Figure 3). When the maize plants’ powder was from plants aged 23 and 43 days (designated young and mature, respectively), interesting differences were revealed. The highest laccase activity was measured when the root powder of mature seedlings was added to the cultures. Similar levels of extracellular laccase were found when young seedlings’ root powder was used instead. However, the addition of mature leaves led to a significant reduction in laccase activity (*p* < 0.05) compared to the addition of roots (mature or young). Leaves from 23-day-old plants induced extracellular laccase activity similar to leaves from 43-day-old plants (with no significant difference).

In the second experiment set, detached seedling roots from various plant species were used in vitro to identify the *M. maydis* ability to exploit them as a nutrition source and to measure variations in laccase secretion during growth. Indeed, the pathogen reacted differently to young roots from maize and other plant species used as field crops in the Hula Valley (Galilee area, northern Israel). The appearance of an infection thread (fungus filaments) inside the roots examined is in the following descending order: barley (where the longest thread was detected, Figure 4), watermelon, maize, cotton, and peanuts. The infection thread spread in all the plant roots tested except for peanuts. On day 3 from inoculation, the barley and watermelon treatments differed significantly (*p* < 0.05) from the rest of the species tested. On day 6 in the barley treatment, the infection thread was significantly (*p* < 0.05) longer compared to all other treatments. At the same time, maize and watermelon reached a similar result that was significantly higher than the cotton result. No infection thread was identified in the negative control treatments (non-inoculated roots). 

In qPCR-based diagnostic of the samples made of a root section 0.5 cm from the infection site; *M. maydis* relative DNA levels were in the following descending order: watermelon, barley, cotton, and corn (Figure 5A). Similar to the measurement of the infection thread that resulted in zero values in the peanut detached roots (Figure 4), no DNA of the pathogen could be measured in those roots. Although *M. maydis* DNA levels in maize roots were less than a third of those in watermelon roots, the incidence (infection percentage) in maize reached 100%, while only 67% was recorded in watermelon (Figure 5B).

The root powder of the above-inspected plant species in MM-submerged *M. maydis* cultures revealed curious differences in fungal biomass and extracellular laccase secretion. Any of the root powder inspected enhanced the fungal growth in comparison to the control (pure minimal medium). Interestingly, it was found that the roots of maize and barley caused a high elevation (more than two-fold) in the pathogen’s wet-biomass (*p* < 0.05, Figure 6A). The cotton root powder treatment had a similar fungal fresh weight as the control but a significant (*p* < 0.05) increase in fungal dry weight. The highest dry weight was recorded in the barley-enriched medium, with a significant difference (*p* < 0.05) from maize and cotton media (Figure 6B). Still, these two enriched media yielded a significantly higher fungal dry weight than in the peanut- and watermelon-enriched media. Upon examining extracellular laccase activity through ABTS oxidation (Figure 6C), the strongest activity was measured by the influence of peanut and watermelon root powder (*p* < 0.05). Maize- and cotton-enriched media, induced relatively moderate laccase activity. The maize medium induced significantly higher laccase activity than the barley-containing medium (*p* < 0.05), which was the least inducing medium of all plant-supplemented media inspected. All laccase activity levels measured in this experiment were relatively low in comparison to the levels measured in Figure 3. This was the result of the addition of 0.1% glucose that led to catabolite repression [32]. In contrast, this glucose addition enables higher fungal-fresh-biomass (compared to Figure 2).

## 4. Discussion

*M. maydis* is considered exotic and unfamiliar in most parts of the world. It has been officially reported so far in only eight countries. Most of the work done on this pathogen has focused on means to restrict its distribution, to control its damages, and to improve our ability to detect and monitor it. Hence, the basic information regarding the biology of the pathogen and its interactions with the host plant remains unknown. This pioneering work provides preliminary results that could encourage the future establishment and widen the database that we have on this pathogen. Such vital information, yet to be achieved, includes the pathogen enzymatic array secreted during pathogenesis, the potential toxins it may produce, and the cellular network of signal transduction pathways that coordinated and synchronized these events. In this context, the laccase enzyme studied here exemplifies the importance of such a database for future coping with this emerging threat.

The laccase enzyme is secreted by fungi and can catalyze the oxidation of phenolic substrates for diverse functions, including virulence factors [28]. However, no information can be found about the role of laccases in the interaction of *M. maydis* with its host plant. The gene for *M. maydis* laccase was identified in our laboratory [33], but its specific roles under various conditions still remain unknown. In this study, we tested the influence of tissues (roots or leaves) from seedlings of different ages on *M. maydis* saprotrophic growth and laccase secretion. The results show basal and very low laccase secretion in minimal medium without glucose or with the addition of 0.1% glucose. The laccase levels increased significantly as corn parts were added as a form of ground tissue. Furthermore, mature roots were found to stimulate laccase secretion more than leaves (mature or young). Thus, it is possible that the composition of mature roots in maize, which contains more hard tissues (such as lignin) compared to leaves, may cause increased secretion of the enzyme, as was found by others [40].

To support this hypothesis, an opposite correlation was found between fungal saprotrophic growth (expressed as biomass) and laccase secretion. For example, in the root powder-supplemented medium, the fungal fresh weight was lower than in the leaf powder-containing medium (Figure 2). Similarly, the peanut root powder, which was the most potent laccase inducer among the plant species inspected here (Figure 6C), was the lesser growth enhancer (accessible nutrition source) (Figure 6A,B). This was also reflected in the fungal inability to develop inside peanut-detached roots (Figure 4). Contrary to this, barley roots-enriched media had the highest fungal biomass in submerged cultures and the longest infection thread in the detached root assay but the lowest extracellular laccase activity. Thus, the difficulty of the fungus to exploit some tissues may be correlated to laccase activity. It should also be taken into consideration that other factors, such as plant-derived inhibitors, may affect these findings.

These findings encourage further examination of host–pathogen interactions and foster a deeper understanding as to under what conditions the fungi use this enzyme. Reviewing the literature suggests that laccase secretion was found to rely on growth conditions and media nutrition composition (reviewed by Reference [41]). Laccase production in fungi is generally affected by different sources of carbon and by the carbon-to-nitrogen ratio (C/N) [42]. Also, laccases were generally secreted at low concentrations, but with the addition of various xenobiotic compounds, such as xylidene, lignin, and veratryl alcohol, fungal laccase production was increased [31]. Some of these compounds affect growth rate or metabolism, while others, such as ethanol, indirectly induce laccase secretion.

The excessive concentrations of sucrose or glucose in many cases reduce laccase production. However, in the cellulose-containing medium, the laccase was produced more efficiently [32]. Supporting that is our observation that the use of the minimal medium with the addition of 0.1% glucose resulted in relatively lower extracellular laccase activity (Figure 6). An interesting example is the case of *Heterobasidion annosum* s.s. and its host plant, the Scots pine [43]. Higher laccase activities were measured in culture growth media comprising sucrose, cellulose, or glucose in comparison to that of cellobiose as a sole carbon source. The severest Scots pine seedlings’ death was observed when they were infected by *H. annosum* s.s. on an extra carbon source such as glucose, suggesting that glucose plays an essential role in the upregulation of laccase activity in this fungus.

Based on the current work’s findings, a subsequent study could examine which additional substances and conditions in seedlings and mature plants induce laccase secretion in *M. maydis*. It would also be very interesting to map the laccase profile in this fungus and to study its expression patterns and role in the pathogen’s interactions with different hosts. An example of such a relationship is the fungal defense against antifungal phenols secreted by the plant. The first step conducted here of adjusting a quantitative/qualitative and rapid test for the secretion of extracellular laccases in the maize pathogen *M. maydis* provides a new research direction for understanding this fungus biology. This and other methods could be used to explore these extracellular enzymes and to enrich our knowledge about them in this unique pathogen.

## 5. Conclusions

The maize late wilt causal agent, *Magnaporthiopsis maydis*, is an emerging pathogen of destructive potential, but very limited information exists about its host relationship. To the best of our knowledge, this work is the first to study *M. maydis* laccases. Previous studies have shown that laccase is secreted during the penetration of pathogenic fungus into the plant to decompose phenols in the cell wall. The current work aimed at studying the effect of the host tissue (roots and leaves at different ages) on *M. maydis* saprophytic growth and secretion of extracellular enzyme laccase. We also studied the effect of roots from various crops on these measures. The results reveal an interesting negative correlation between the ability of the fungus to grow on different plant tissues and the extracellular laccase it produces. These conflicting effects hint at the role of laccase in exploiting different nutrition sources. The results of this study are an important step towards studying *M. maydis* host–pathogen interactions under different conditions. The findings expand the scientific research tools currently available for studying this pathogen’s behavior and for exploring its laccase involvement in pathogenesis.

## Figures and Tables

**Figure 1 jof-06-00063-f001:**
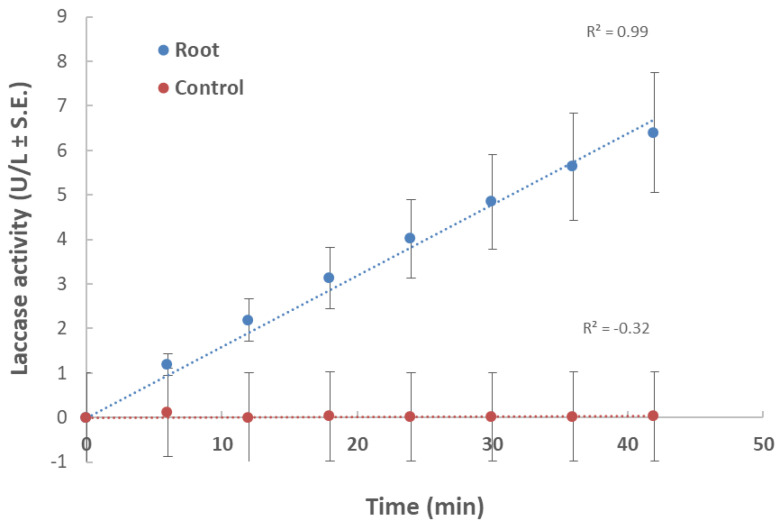
Effect of maize root powder on *M. maydis* extracellular laccase activity over time: The fungus was grown for seven days in 50 mL of a minimal medium (MM) with 0.1% glucose to which 750 mg of shredded plant parts in liquid nitrogen from 40-day-old Prelude cv. maize plant was added. The control is MM with *M. maydis*, without root powder. To the growth medium of each treatment and the control (after purification), 2,2′-azinobis (3-ethylbenzothiazoline-6-sulfonic acid) (ABTS) was added. The enzymatic units (U) were defined as the amount of enzyme oxidizing 1 µmol ABTS in 1 min at 28 ± 1 °C in the dark. The values shown are an average of five repetitions.

**Figure 2 jof-06-00063-f002:**
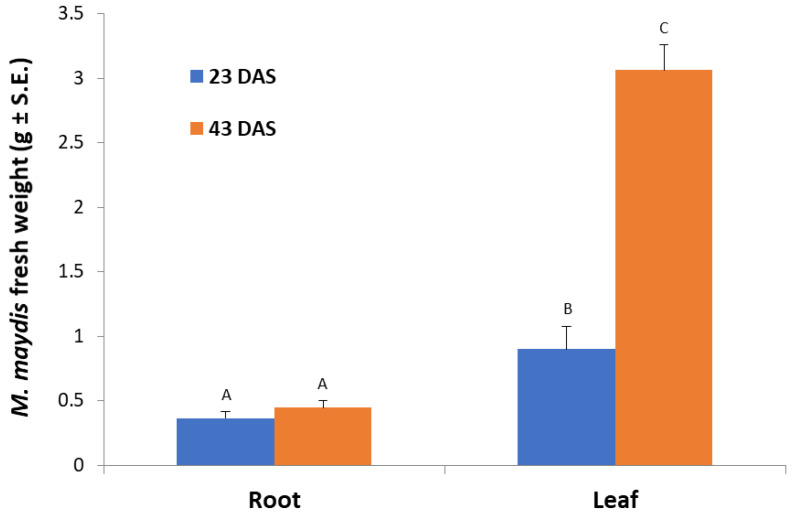
Effect of maize roots and leaves from plants at different ages on *M. maydis* growth: Corn plants from the Jubilee cv. were grown in a growth chamber for 23 or 43 days from sowing (DAS). The plants’ parts powder was added to submerged *M. maydis* cultures in MM without glucose. Each repeat consists of the entire roots or leaves of one plant. The values represent the average mycelia fresh weight of six repetitions obtained after seven days of incubation. Different letters (A, B, and C) mark a significant difference (*p* < 0.05, ANOVA).

**Figure 3 jof-06-00063-f003:**
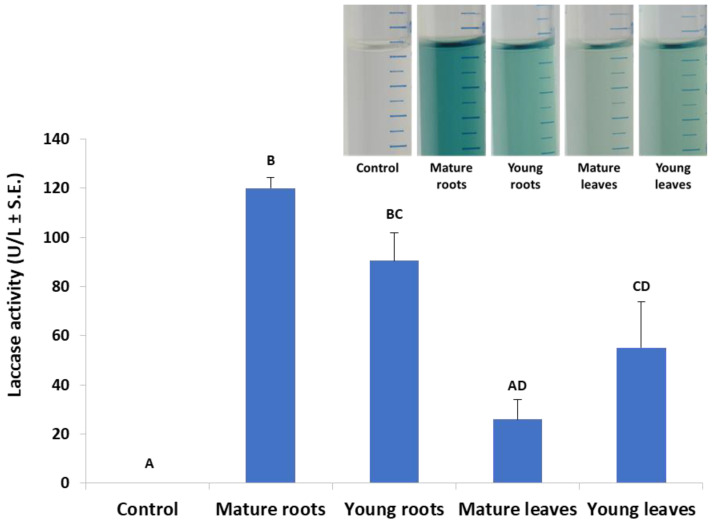
Effect of plant age and organ (roots or leaves) on *M. maydis* extracellular laccase activity: The laccase test was performed, as described in the legend of Figure 1. The experiment conditions are described in the legend of Figure 2. Maize plants’ powder from plants aged 23 and 43 days (designated young and mature, respectively) was added to MM without glucose. The resulting growth medium was incubated with ABTS for 10 min. The enzymatic units (U) were defined as the amount of enzyme oxidizing 1 µmol ABTS in 1 min at 28 ± 1 °C in the dark. The values represent an average of 4–6 repetitions with the exception of the control (minimal medium only), which included two repetitions. Each repeat consists of the entire roots or leaves of one plant. Levels not connected by the same letter (A, B, C, and D) are significantly different (*p* < 0.05, ANOVA). The insert is a representative photo of the experiment treatments after 30 min of incubation.

**Figure 4 jof-06-00063-f004:**
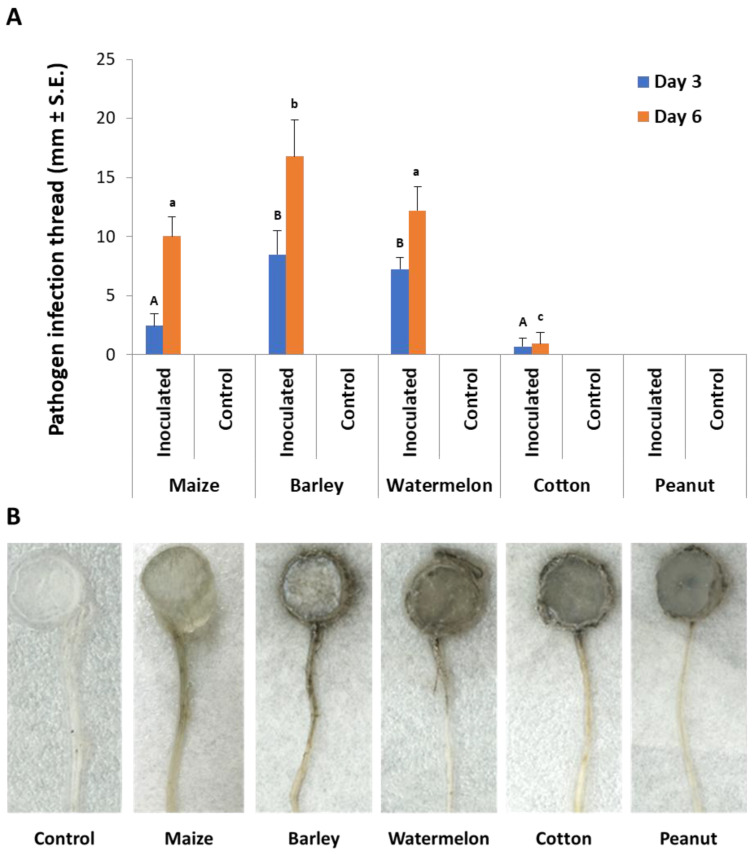
Detection of *M. maydis* in infected detached roots of different plant species: (**A**) The average length of infection threads (fungus filaments) within the root. Maize plant roots were from Prelude cv. Each column is an average of five replications. The error lines represent a standard error. Levels not connected by the same letter (A and B—3 days; a, b, and c—6 days) are significantly different (*p* < 0.05, ANOVA). (**B**) Photograph of representative detached roots after six days: The infection threads are seen as a dark stripe developing along the root from the culture disc.

**Figure 5 jof-06-00063-f005:**
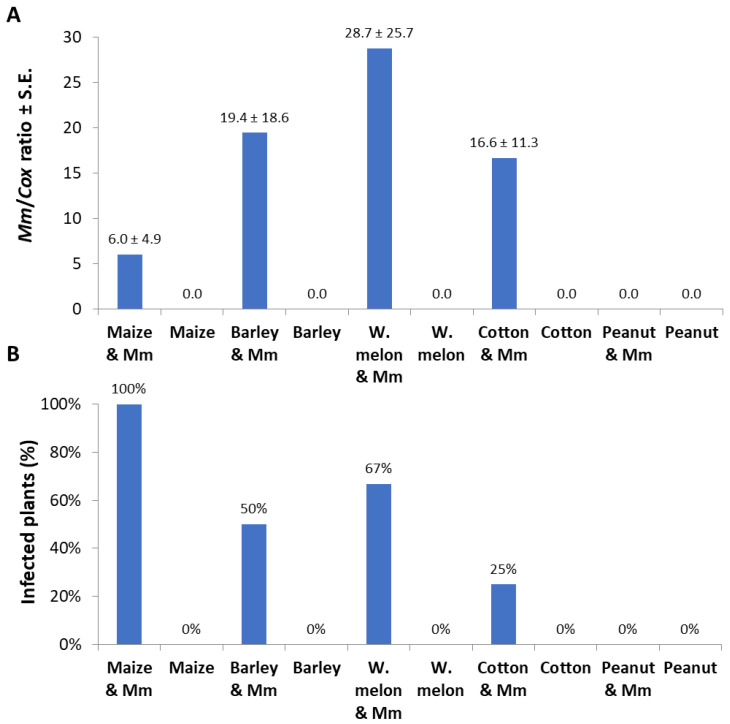
Detached roots pathogenicity assay quantitative real-time PCR results: The experiment is described in Figure 4. The qPCR method used is very sensitive, and some of the measurements (repeats) resulted in zero values because of an objective difficulty in achieving high and uniform inoculation with this particular pathogen Israeli strains. Therefore, two measures to describe the results were used: pathogen colonization degree (mean relative value) and incidence (percentages of infected plants that had positive detection using the qPCR method). (**A**) Pathogen colonization degree: *M. maydis* relative DNA (*Mm*) levels normalized to the housekeeping gene cytochrome c oxidase (*Cox*) DNA. (**B**) Incidence: rate of infected plants (with *M. maydis* DNA inside their tissues) in percentage, identified by the qPCR. Controls—noninfected plants tested in these experiments (all had zero levels of *M. maydis* relative DNA). Values represent the average of five replications ± standard error.

**Figure 6 jof-06-00063-f006:**
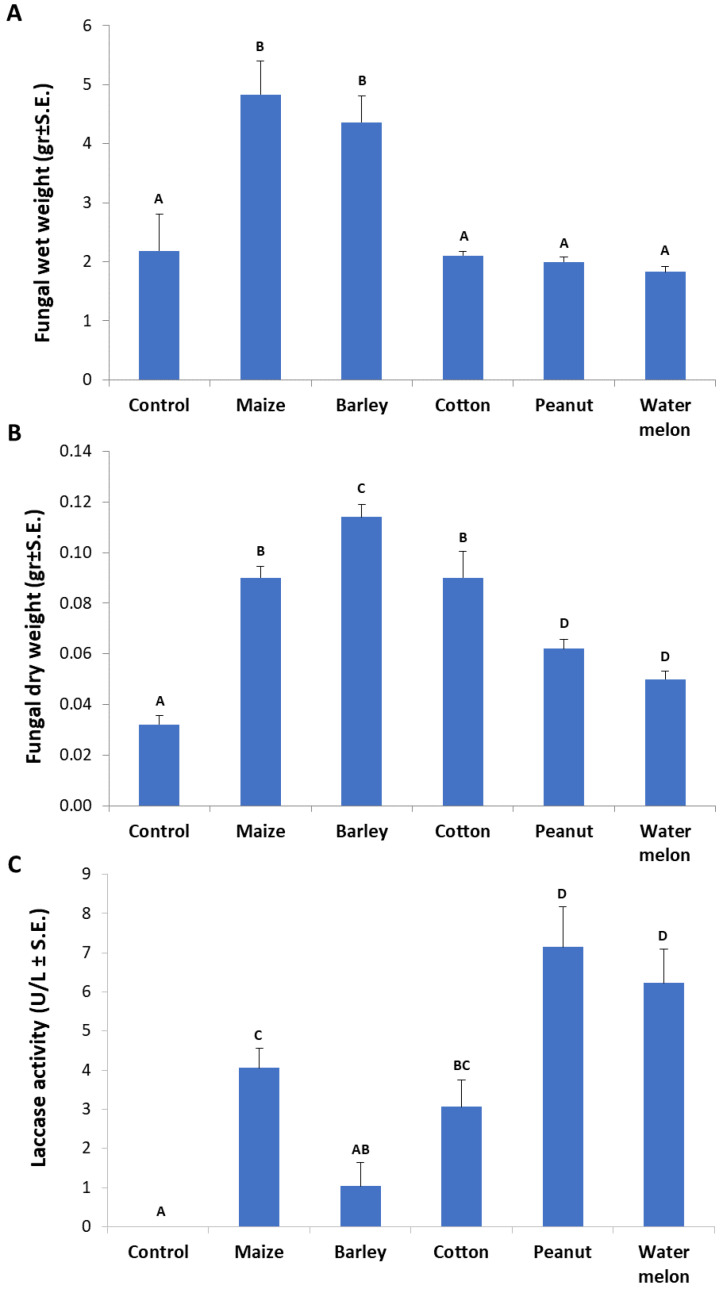
Influence of various plant species root powder on *M. maydis* growth in liquid minimal medium: The fungus was grown for seven days in 50 mL of a minimal medium with 0.1% glucose and 750 mg of ground plant parts from 40-day-old Prelude cv. maize plants and other plant species. (**A**) Fungal wet weight. (**B**) Fungal dry weight. (**C**) Laccase assay (performed as described in Figure 1). The enzymatic units (U) were defined as the amount of enzyme oxidizing 1 µmol ABTS in 1 min at 28 ± 1 °C in the dark. Bars indicate the mean of five replications. Levels not connected by the same letter (A, B, C, and D) are significantly different (*p* < 0.05, ANOVA).

**Table 1 jof-06-00063-t001:** Primers for *Magnaporthiopsis maydis* qPCR detection. ^1^

Pairs	Primer	Sequence	Uses	Amplification	References
**Pair 1**	A200a-for A200a-rev	5′-CCGACGCCTAAAATACAGGA-3′ 5′-GGGCTTTTTAGGGCCTTTTT-3′	Target gene	*M maydis* AFLP-derived species-specific fragment	[24]
**Pair 2**	COX-F COX-R	5′-GTATGCCACGTCGCATTCCAGA-3′ 5′-CAACTACGGATATATAAGRRCCRRAACTG-3′	Control	Cytochrome c oxidase (COX) gene product	[38,39]

^1^ qPCR—quantitative real-time PCR. AFLP—amplified fragment length polymorphism.

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
