# Peer review of "Potential Role of Laccases in the Relationship of the Maize Late Wilt Causal Agent, Magnaporthiopsis maydis, and Its Host"

_jof, 2020, doi:10.3390/jof6020063_

Round 1

Reviewer 1 Report

Interesting research subject, but some methods are incorrect.

Laccase activity cannot be measured as described in the manuscript. The results obtained by such method are incorrect. And laccase activity should be presented in Units, not just as optical density change.

Results presented at figure 5 are very confusing. Standard error is too high to make any conclusions. Perhaps, more replications should be done.

Some parts of the manuscript are written in very unclear manner (for example, L257-259, L301-303, L331-332).

As mentioned, the laccase activity measurement method is not correct; hence, almost all conclusions are not supported by the results. Also, I should note that the purpose of the development of a new technique for laccase activity measurement is unclear. There are a lot of papers where laccase activity measurement using ABTS is described. For example, see Eggert et al. 1996, Gianfreda et al. 1998 or Schlosser et al. 1997.

Author Response

Responses to Reviewer 1’s comments

We thank the reviewer for investing substantial work that contributes significantly to this manuscript. The remarks and suggestions improved this scientific paper remarkably and made it more accurate, clear, focused, and well-structured. Your contribution is greatly appreciated.

General comments:

Laccase activity cannot be measured as described in the manuscript. The results obtained by such a method are incorrect. And laccase activity should be presented in Units, not just as optical density change.

The reviewer is correct. Therefore, we recalculated and replaced all the laccase absorbance values to activity units. We also replaced all the laccase activity figures (Figures 1, 3, 6C) and the relevant text. We also redid all statistical analyses of the new data. The laccase activity assay description (Section 2.3) in Materials and Methods was corrected and rephrased as follows: “The enzymatic units (U) defined as the amount of enzyme oxidizing 1 µmol ABTS in 1 min and calculated by the following formula: ?/? = (∆? × ?? × 10^6) / (? × ? × ??) [36,37], where, ∆A = change in absorbance per min; Vt = total volume measured; 106 = correction factor; Vs = volume of enzyme; (ε) of ABTS = Extinction coefficient for the oxidation that is 36000 M-1 cm-1; d = path length of the optical cell is 1 cm.” (lines 143-147)

Despite this correction to the data, the laccase assay main findings remained unchanged, thus the interpretation of the results remains as before (with some minor adjustments).

References: 

  1. Baltierra-Trejo, E.; Márquez-Benavides, L.; Sánchez-Yáñez, J.M. Inconsistencies and ambiguities in calculating enzyme activity: the case of laccase. Journal of Microbiological Methods 2015, 119, 126-131.
  2. Agrawal, K.; Verma, P. Laccase: addressing the ambivalence associated with the calculation of enzyme activity. Biotech 2019, 9, 365.

Results presented in Figure 5 are very confusing. Standard error is too high to make any conclusions. Perhaps more replications should be done.

The reviewer is correct; this is indeed an important issue that should be clarified.

The qPCR method we used is very sensitive and capable of detecting variations in the amount of the pathogens’ DNA inside the host plant tissues with a million-fold difference (see, for example, Degani et al., PloS One 2018, 13, e0208353). Moreover, in pathogenicity trials, some of the measurements (repeats) resulted in zero values due to an objective difficulty in achieving high and uniform inoculation with this particular pathogen Israeli strains (see Degani et al., Plant Disease, 2019, 103, 238-248).

Since the mean of the results is affected by extreme values, we use two measures to describe the results: severity (mean relative value) and prevalence (percentages of infected plants that had positive detection using our qPCR method). This approach is crucial for an accurate description and a better understanding of the results.

The following explanation was added to Figure 5 footnotes: “The qPCR method used is very sensitive and some of the measurements (repeats) resulted in zero values because of an objective difficulty in achieving high and uniform inoculation with this particular pathogen of Israeli strains. Therefore, two measures to describe the results were used: severity (mean relative value) and prevalence (percentages of infected plants that had positive detection using the qPCR method).” (lines 267-271)

Some parts of the manuscript are written in a very unclear manner (for example, L257-259, L301-303, L331-332).

The reviewer is correct; these sections were indeed unclear. Therefore, we rewrote all of them, as suggested.

Lines 276-281 (previously 257-259) now reads: “The root powder of the above-inspected plant species in MM submerged M. maydis cultures revealed curious differences in fungal biomass and extracellular laccase secretion. Any of the root-powder inspected enhanced the fungal growth in comparison to the control (pure minimal medium). Interestingly it was found that the roots of maize and barley caused a high elevation (more than two-fold) in the pathogen’s wet-biomass (p < 0.05, Figure 6A). The cotton root powder treatment had a similar fungal fresh weight as the control, but a significant (p < 0.05) increase in fungal dry weight.”

Lines 338-340 (previously 301-303) now reads: “Also, laccases were generally secreted at low concentrations, but with the addition of various xenobiotic compounds, such as xylidene, lignin, and veratryl alcohol, fungal laccase production was increased.”

Lines 368-369 (previously 331-332) now reads: “These conflicting effects hint at the role of laccase in exploiting different nutrition sources.”

As mentioned, the laccase activity measurement method is not correct; hence, almost all conclusions are not supported by the results.

As detailed in our response above, in the general comments section, we altered all the absorbance values to laccase activity values expressed in U/L. Nevertheless, this change caused only minor changes in the values tendencies or the statistical differences between treatments. Therefore, the conclusions remain unchanged and are supported by the results.

Also, I should note that the purpose of the development of a new technique for laccase activity measurement is unclear. There are a lot of papers where laccase activity measurement using ABTS is described. For example, see Eggert et al. 1996, Gianfreda et al. 1998 or Schlosser et al. 1997.

Indeed, this is a correct remark, the closing paragraph at the end of the discussion was edited, and the term “new research tool” was rephrased. The paragraph now reads: “The first step conducted here of adjusting a quantitative/qualitative and rapid test for the secretion of extracellular laccases in the maize pathogen M. maydis provides a new research direction for understanding this fungus biology. This and other methods could be used to explore these extracellular enzymes and enrich our knowledge about them in this unique pathogen.” (lines 354-358)

Reviewer 2 Report

Dear authors,

Potential role of laccases in the relationship of the maize late wilt causal agent, Magnaporthiopsis maydis, and its host is well-written manuscript and the topic is relevant and interesting.  Authors attempted to use in vitro growth media conditions and root pathogenicity assay and measured variations in fungal growth weight, DNA levels, and laccase activity of the fungus. However, this is just preliminary study and authors need to further study the association of laccase activity with growth in M. maydis. Authors should add more discussion based on their observations in this study.  

There remain some issues that do not allow your paper to be accepted for publication at this time and authors should consider the comments useful for further revision of the manuscript. Importantly, there are lot of grammatical errors that authors should correct before submitting. Additionally, the quality of the figures is not very good and the images are blurred.

Minor comments:

Line 87:         The gene for laccase should be changed to gene encoding laccase

Line 87-90:    The sentence is not clear. Please rewrite this sentence.

Line 100:       change the word confirm to confirmed

Line 141:       change the word to using a hume hood

Line 108:       change rounds per minute to revolutions per minute

Line 126-129:  ABTS, please discuss this in detail. The original product in the Sigma website, the readings is at 405 nm. Authors however use 450 nm. Is there any particular reason to change the wavelength? Also what authors mean by mm ABTS ?  The unit is not very clear.

Line 182-184:  difference from the leaves-containing medium was eradicated. Change the term eradicated- its not the right term.

Fig.1 . – The legend MM(-Hm2) is confusing. Please make sure to use the terms consistently

Fig.3 -  Authors indicate that each repeat of experiment consists of the entire roots or leaves of one plant. However, if there are more roots or leaves in one plant compared to others, do authors see any difference in color in the Laccase assay ?  For the experimental accuracy, authors should normalize the quantity of the roots and leaves from each plant.

Author Response

Responses to Reviewer 2’s comments

We would like to express our sincere appreciation to the reviewer for important and helpful suggestions and advice. The time and effort invested are greatly appreciated, and without a doubt, contributed to the manuscript and significantly improved it. Thank you.

General comments:

This is just preliminary study and authors need to further study the association of laccase activity with growth in M. maydis. Authors should add more discussion based on their observations in this study. 

We agree with the reviewer. M. maydis is considered to be exotic and unfamiliar in most parts of the world. It has been officially reported (in peer-reviewed scientific papers) so far in only eight countries. Most of the work done on this pathogen has focused on means to restrict its distribution, control its damages, and improve our ability to detect and monitor it. Hence, the basic information regarding the biology of the pathogen and its interactions with the host plant remains unknown. This pioneering work provides preliminary results that could encourage the future establishment and widening the database that we have on this pathogen. Such vital information, yet to achieve, includes the pathogen enzymatic array secreted during pathogenesis, the potential toxins it may produce, and the cellular network of signal transduction pathways that coordinated and synchronized these events. In this context, the laccase enzyme studied here exemplifies the importance of such a database for future coping with this emerging threat.

This explanation was added to the discussion (lines 302-311).

There are lot of grammatical errors that authors should correct before submitting.

The revised manuscript was edited by a professional English scientific copy editor.

Additionally, the quality of the figures is not very good and the images are blurred.

Perhaps transforming the document to PDF format caused the low-resolution appearance of some of the figures. All figures were embedded in the text in high resolution, and it is best to view them in the MS-Word format manuscript version.

Minor corrections:

Line 87: The gene for laccase should be changed to gene encoding laccase

Corrected as advised.

Lines 87-90: The sentence is not clear. Please rewrite this sentence.

The sentence was rephrased and now reads: “The gene encoding laccase in M. maydis was previously identified and sequenced in our laboratory [33]. Still, it is unknown under what conditions the enzyme is produced and secreted, and for what purpose (fungal development or interactions with the host plant).” (Lines 87-89)

Line 100: change the word confirm to confirmed

Corrected as advised.

Line 141: change the word to using a hume hood

The term “Fume hood” is correct. See, for example, https://en.wikipedia.org/wiki/Fume_hood.

Line 108:  change rounds per minute to revolutions per minute

Corrected as advised.

Line 126-129:  ABTS, please discuss this in detail. The original product in the Sigma website, the reading is at 405 nm. Authors however use 450 nm. Is there any particular reason to change the wavelength?

Indeed as written in the Sigma-Aldrich website, the ABTS substrate produces a soluble end product that is green in color and can be read spectrophotometrically at 405 nm. However, there are different wavelengths described in the literature regarding the laccase assay.  For example, laccase activity was determined spectrophotometrically at 420 nm [1], 436 nm [2] and 450 nm [3]. Reviewing the literature led to the conclusion that the majority of work was done with 420 nm, which is today considered the optimal wavelength for the ABTS substrate. Thus, although our work was conducted near this optimum wavelength (at 450 nm), the measurements were still strong enough and differed significantly from each other (p<0.05), and allowed an accurate and detailed understanding of the host tissue effect on M. maydis extracellular laccase activity.

The following explanation was added to the laccase activity assay description (Section 2.3) in Materials and Methods (lines 129-134): “Today, the majority of works described in the literature are done with 420 nm, which is considered the optimal wavelength for the ABTS substrate. Thus, although the current work was conducted near this optimum wavelength (at 450 nm), the measurements were still strong enough and differed significantly from each other (p < 0.05) to allow an accurate and detailed understanding of the host tissue effect on M. maydis extracellular laccase activity.“

References:

  1. Agrawal, K.; Verma, P. Laccase: addressing the ambivalence associated with the calculation of enzyme activity. Biotech 2019, 9, 365.
  2. Sheikhi, F.; Roayaei Ardakani, M.; Enayatizamir, N.; Rodriguez-Couto, S. The determination of assay for laccase of Bacillus subtilis wpi with two classes of chemical compounds as substrates. Indian J Microbiol 2012, 52, 701-707.
  3. Hofrichter, M.; Fritsche, W.J. Depolymerization of low-rank coal by extracellular fungal enzyme systems. II. The ligninolytic enzymes of the coal-humic-acid-depolymerizing fungus Nematoloma frowardii b19. Biotechnology 1997, 47, 419-424.

Also, what do the authors mean by mm ABTS?  The unit is not very clear.

The ABTS concentration units were corrected to “50 mM.” This was just a typo.

Line 182-184:  difference from the leaves-containing medium was eradicated. Change the term eradicated - its not the right term.

The term “eradicated” was replaced with the word “minimized.” The sentence now reads: “Higher activities of laccase were measured in culture medium containing roots after 10 min of incubation (not statistically different from the roots), but at 30 min onwards, this difference from the leaves-containing medium was minimized.” (lines 193-195)

Fig. 1 . – The legend MM(-Hm2) is confusing. Please make sure to use the terms consistently.

The reviewer is correct. Therefore, we updated all legends regarding the controls. The figures legends now include two terms: Control – minimal medium (MM) with M. maydis, and MM – minimal medium only.

Fig. 3 -  Authors indicate that each repeat of experiment consists of the entire roots or leaves of one plant. However, if there are more roots or leaves in one plant compared to others, do authors see any difference in color in the laccase assay ? 

Indeed, natural phenotypic variations exist between the plant samples (repeats) within each treatment. However, these differences are minor, as reflected by the standard error bars in the laccase assay.   

For experimental accuracy, authors should normalize the quantity of the roots and leaves from each plant.

We did normalize the quantity of the roots and leaves from each plant. As stated in Materials and Methods: “To these flasks, 750 mg of plant parts (roots or leaves) powder (prepared by grinding in liquid nitrogen) was added.” (lines 107-108)

Reviewer 3 Report

The article is very well written, with clear objectives, and the work proposal is very well founded and of great economic and social importance, as it seeks solutions to a disease that threatens the production of an important food product, which is corn.

I have no observations to make about the abstract, introduction, and discussion of the results, as the wording is very clear and concise.

The bibliography is, for the most part, out of date, but this is offset by the practical nature of the article, the results obtained, and the good exploration that the authors have made of such results. However, I suggest updating the bibliography.

Items to be corrected:

Materials and Methods

  • Lines 107-111: “To these flasks, 750 mg of plant parts powder (prepared by grinding in liquid nitrogen) was added. Cultures grown at 28±1°C in complete darkness on a rotary shaker at 150 rounds per minute (RPM) for one week. At the end of the growth, the fungus and plant components were filtered from the substrate through double-layer Whatman 3 mm filter papers in a Büchner funnel, dried at 40â—‹C for 24 hours, and weighed. “

- The drying of the mycelial mass and plant components for 24 hours at 40 ° C is not sufficient for the complete dehydration of the biological material. This procedure must be repeated until a constant weight is obtained (around 2 or 3 days)

Results

  • Figure 1 – page 5

Explain in the caption what MM and MM (-HM2) mean, as these initials are mentioned in Material and Methods in the middle of the text.

- I did not detect the initials, in the graph, of the activity of the MM samples (-HM2). Have they been plotted?

Page 8, Line 263: “... examining extracellular laccase activity through ABTS dismantling (Figure 6B), the strongest activity….”

- Substituir “Figure B” por “Figure C”

Page 9 – Figure 6: Correct the caption, because Figure A was mentioned and C was called B. Figure B was not mentioned.

Author Response

Responses to Reviewer 3’s comments

We thank the reviewer for investing time and effort, which contributed to this manuscript. The helpful and necessary remarks and suggestions improved this scientific paper and made it more accurate, clear and focused. Thank you.

General comments:

The bibliography is, for the most part, out of date, but this is offset by the practical nature of the article, the results obtained, and the good exploration that the authors have made of such results. However, I suggest updating the bibliography.

This is a correct remark. According to the reviewer suggestion, we updated the bibliography and added the following citations:

  1. Baltierra-Trejo, E.; Márquez-Benavides, L.; Sánchez-Yáñez, J.M. Inconsistencies and ambiguities in calculating enzyme activity: the case of laccase. Journal of Microbiological Methods 2015, 119, 126-131.
  2. Agrawal, K.; Verma, P. Laccase: addressing the ambivalence associated with the calculation of enzyme activity. Biotech 2019, 9, 365.
  3. Patel, A.; Patel, V.; Patel, R.; Trivedi, U.; Patel, K.J. Fungal laccases: versatile green catalyst for bioremediation of organopollutants. Emerging Technologies in Environmental Bioremediation 2020, 85-129.
  4. Rodríguez-Couto, S. Fungal laccase: a versatile enzyme for biotechnological applications. In Recent advancement in white biotechnology through fungi, Springer: 2019; pp 429-457.
  5. Janusz, G.; Pawlik, A.; Świderska-Burek, U.; Polak, J.; Sulej, J.; Jarosz-Wilkołazka, A.; Paszczyński, A.J. Laccase properties, physiological functions, and evolution. International Journal of Molecular Sciences 2020, 21, 966.
  6. Singh, D.; Gupta, N.J.B. Microbial laccase: A robust enzyme and its industrial applications. Biologia 2020, 1-11.

Specific comments:

Lines 107-111: “To these flasks, 750 mg of plant parts (roots or leaves) powder (prepared by grinding in liquid nitrogen) was added. Cultures were grown at 28±1°C in complete darkness on a rotary shaker at 150 revolutions per minute (RPM) for one week. At the end of the growth, the fungus and plant components were filtered from the substrate through double-layer Whatman 3 mm filter papers in a Büchner funnel, dried at 40â—‹C for 24 hours to nearly complete dryness and weighed“

- The drying of the mycelial mass and plant components for 24 hours at 40°C is not sufficient for the complete dehydration of the biological material. This procedure must be repeated until a constant weight is obtained (around 2 or 3 days)

We agree this is indeed a good recommendation. Ufortenetly we can not repeat these measurements. Thous the following correction was made to the text (line 109-111): “At the end of the growth, the fungus and plant components were filtered from the substrate through double-layer Whatman 3 mm filter papers in a Büchner funnel, dried at 40â—‹C for 24 hours to a nearly complete dryness and weighed.”

Figure 1 – page 5

Explain in the caption what MM and MM (-HM2) mean, as these initials are mentioned in Material and Methods in the middle of the text.

- I did not detect the initials, in the graph, of the activity of the MM samples (-HM2). Have they been plotted?

The reviewer is correct. Therefore, we updated all legends regarding the controls. The figure legends are now include two terms: Control – minimal medium (MM) with M. maydis, and MM – minimal medium only.

Also note that, as suggested by Reviewer 1, Figure 1 was replaced, and the laccase absorbance values were replaced with activity units.

The MM treatment has been plotted in Figure 1, but it has similar values to the control treatment. Therefore the marks of both treatments are at the same locations. 

Page 8, Line 263: “... examining extracellular laccase activity through ABTS dismantling (Figure 6B), the strongest activity….”

- Substitute “Figure B” with “Figure C”

Corrected as advised.

Page 9 – Figure 6: Correct the caption, because Figure A was mentioned and C was called B. Figure B was not mentioned.

Corrected as advised.

Round 2

Reviewer 1 Report

First of all, I should note, that enzyme activity should be measured via initial velocities. And “the central theme among all enzyme activity measurements is to ensure measurement of initial velocities.” (quoted from Harris, T. K., & Keshwani, M. M. (2009). Chapter 7 Measurement of Enzyme Activity. Guide to Protein Purification, 2nd Edition, 57–71. doi:10.1016/s0076-6879(09)63007-x). As can be seen from presented results, there is no evidences that the initial reaction velocities were measured. So, the enzyme activity cannot be calculated from the presented data. New experiments are required. I highly recommend to measure the absorbance in the kinetic mode to ensure that the initial velocities are measured.  So, I made no commentaries to the L192-L206 part and Figure 1 as it does not make sense in its current form.

I also would like to ask the authors read the manuscript again carefully as there are some missing words, incorrect references etc.

L17: I suppose “under what conditions it is expressed”?

L77: low molecular weight..? I suppose, the word “compound” is missing.

L138: please, add the concentration of buffer.

L148, 162: please, check the italics.

L154: maybe, “cut from the growing edge”?

L207 and further: please, concern using  the word “supplemented” instead of “embedded”.

L214 and further: please, concern using one name: shoots or leaves. It is a little bit confusing when different words are used.  

L216: I suppose that mm means MM, minimal media?

L218: “weight” not “wet”, I suppose?

L224-227, 227-229 and 320-321: in both cases you cannot say that one value is higher than another while there in no significant difference. This should be rephrased.

L232: maybe, “in legend of Figure 2”?

L285: the term “dismantling” cannot be used for ABTS there. “Oxidation” will be more correct.

L321-323, L313 and L337: please, check the references. I cannot find the corresponding information in these publications.

L328: I’m not sure that the accessibility of nutrition source is the only reason. Maybe some plant-derived inhibitors can affect the fungal growth.

L342-349: This paragraph is just review of the literature. Please, add some discussion concerning your results.  

Author Response

Responses to Reviewer 1’s report 2 comments

We would like to express our appreciation to the reviewer for the important and helpful corrections, suggestions, and advice. We are sure that this contribution significantly improved the manuscript. Thank you.

First of all, I should note that enzyme activity should be measured via initial velocities. And “the central theme among all enzyme activity measurements is to ensure measurement of initial velocities” (quoted from Harris, T. K., & Keshwani, M. M. (2009). Chapter 7 Measurement of Enzyme Activity. Guide to Protein Purification, 2nd Edition, 57-71. doi:10.1016/s0076-6879(09)63007-x). As can be seen from the results presented, there is no evidence that the initial reaction velocities were measured. So, enzyme activity cannot be calculated from the data presented. New experiments are required. I highly recommend measuring the absorbance in the kinetic mode to ensure that the initial velocities are measured. So, I made no commentaries to the L192-L206 part and Figure 1 as it does not make sense in its current form.

Please note that the laccase activity was calculated using a well-established formula used for the same purpose as is common worldwide. We reviewed more than 20 publications published in recent years in leading scientific platforms. The laccase assay procedure and the laccase activity calculation carried out in the current work were similar to what was done in most of them. Also, the laccase activity calculation in many of those studies was conducted without measuring or calculating enzymatic-initial velocity (V0). This calculation is used mostly for determining enzyme kinetic properties such as Km, V0max and Kcat in studies dedicated to that purpose.

This being said, we agree with reviewer 1’s comments that the old Figure 1 was not appropriate. Therefore, we added new experiment results to address this concern. Figure 1 was replaced with a new figure that now describes in detail the changes in laccase activity over time expressed as increasing product concentration (U/L). This ensured that the substrate used (ABTS) was at an excessive concentration and was not consumed during this time point, and that measurable differences can be identified with appropriate sensitivity. Also, the curve shows that the selected time point (10 min) is within the linear curve zone.

Indeed measuring laccase activity at this time point (10 min) is the recommended procedure adapted in many works done with ABTS substrate. See, for example:

  1. Agrawal, K., & Verma, P. Laccase: addressing the ambivalence associated with the calculation of enzyme activity. Biotech, (2019), 9, 365.
  2. Góralczyk-BiÅ„kowska, A., et al. Laccase activity of the ascomycete fungus Nectriella pironii and innovative strategies for its production on leaf litter of an urban park. Plos one15(4) (2020), e0231453.‏
  3. Jeon, S-J., & Su-Jin, L. Purification and characterization of the laccase involved in dye decolorization by the white-rot fungus Marasmius scorodonius.  Microbiol. Biotechnol27(6) (2017), 1120-1127.‏
  4. Irfan, M., Mehmood, S., Irshad, M., & Anwar, Z. Optimized production, purification and molecular characterization of fungal laccase through Alternaria alternata. Turkish Journal of Biochemistry43(6) (2018), 613-622.
  5. Odeniyi, O. A., et al. Production characteristics, activity patterns and biodecolourisation applications of thermostable laccases from Corynebacterium efficiens and Enterobacter ludwigii. Journal of Scientific & industrial research 76(09) (2017), 562-569.

We would also like to clarify that the aim of this work was not to study the kinetic properties of M. maydis laccases. Purification and characterization of M. maydis laccase enzymes is an intriguing topic that should be explored in future studies. Instead, this pioneering work on the scope of M. maydis biology and host relationships is meant to study the laccase secretion response in various host-related situations.

The recommendation to measure the absorbance in the kinetic mode to ensure that the initial velocities are measured is correct could be conducted. Still, we think this procedure will require considerable investment in time and effort and will have only a minor contribution to this work’s results and their interpretation. We will elaborate on this to explain our opinion:

This task will require nearly two months to accomplish since we need to grow maize seedlings for 40 days and to produce root powder. This powder should then be added to submerged M. maydis cultures to induce laccase secretion (an additional week). This laccase, after separation from the cultures, could be tested with increasing concentrations of ABTS to produce extracellular laccase activity over time curves. Only then could we extract the curve slopes and create the Michaelis-Menten graph that would allow us to identify the initial velocity (V0) location of the time point (10 min) we used in this work.

So, what is the benefit to this work of finding the V0? We are most certainly not conducting our laccase activity tests within the lag phase — this assumption relies on reviewing many studies carried out in other fungi. Most of them were conducted with the ABTS substrate at 3 min or more (many at 10 min as we did). If the V0 will show that we are in the linear phase of the curve, this is the ideal situation for measuring the laccase kinetic properties, but this data will not affect the results presented here. And finally, if we are in or near the V0max zone, this means that we have the highest activity production per min (Kcat) at the time point we selected. Apparently, we indeed made our measurements at the maximum reaction rate (the V0max zone), because we used a high concentration of the substrate (50 mM). This means that we measured the highest production rate of the fungal extracellular laccase, produced under the influence of the plants’ parts addition.  

Overall, this information, although interesting, will have no impact on the results described in this manuscript or their conclusions. Moreover, the conditions used here enable clear identification of differences between the treatments, and between the treatments and the control, and a high sensitivity that is expressed in the statistically significant differences.

I also would like to ask the authors read the manuscript again carefully as there are some missing words, incorrect references, etc.

As recommended by the reviewer, we carefully read the entire manuscript and made the necessary corrections to ensure that all the data presented, the supporting information, conclusions, references, etc. are accurate and clearly written.

L17: I suppose “under what conditions it is expressed”?

Corrected as advised.

L77: low molecular weight..? I suppose, the word “compound” is missing.

Indeed, the sentence was corrected as advised.

L138: please, add the concentration of buffer.

The buffer citrate concentration was added (50 mM).

L148, 162: please, check the italics.

To the best of our understanding, this is the customary format in this journal. The manuscript formatting was done by the journal team at the final stage of the manuscript production.

L154: maybe, “cut from the growing edge”?

Indeed, corrected as adviced.

L207 and further: please, concern using the word “supplemented” instead of “embedded.”

Corrected as adviced. The word “supplemented” was used instead of “embedded.”

L214 and further: please, concern using one name: shoots or leaves. It is a little bit confusing when different words are used.

Corrected as advised. The word “leaves” was used instead of “shoots.”

L216: I suppose that mm means MM, minimal media?

Indeed, corrected as adviced.

L218: “weight” not “wet”, I suppose?

Indeed, corrected as advised.

L224-227, 227-229 and 320-321: in both cases you cannot say that one value is higher than another while there is no significant difference. This should be rephrased.

The sentences were rephrased as suggested by the reviewer:

Lines 224-229: “The highest laccase activity was measured when the root powder of mature seedlings was added to the cultures. Similar levels of extracellular laccase were found when young seedlings’ root powder was used instead. However, the addition of mature leaves led to a significant reduction in laccase activity (p < 0.05) compared to the addition of roots (mature or young). Leaves from 23-day-old plants induced extracellular laccase activity similar to leaves from 43-day-old plants (with no significant difference).”

Lines 321-324: “Furthermore, mature roots were found to stimulate laccase secretion more than leaves (mature or young). Thus, it is possible that the composition of mature roots in maize, which contains more hard tissues (such as lignin) compared to leaves, may cause increased secretion of the enzyme, as was found by others [40].”

L232: maybe, “in legend of Figure 2”?

Corrected as advised.

L285: the term “dismantling” cannot be used for ABTS there. “Oxidation” will be more correct.

Indeed, corrected as advised.

L321-323, L313 and L337: please, check the references. I cannot find the corresponding information in these publications.

Line 314 – Reference 28 is correct. Please see it in this link:

https://academic.oup.com/femsre/article/30/2/215/2367606

Line 324 – Reference 40 is correct. Please see it in this link:

https://pubs.acs.org/doi/abs/10.1021/cr000115l

Line 339 – Reference 41 is correct. Please see it in this link:

L328: I’m not sure that the accessibility of nutrition source is the only reason. Maybe some plant-derived inhibitors can affect the fungal growth.

This is a correct remark. The closing sentence at the end of the paragraph was rephrased, as suggested, so the conclusion presented would be drafted more carefully: “Thus, the difficulty of the fungus to exploit some tissues may be correlated to laccase activity. It should also be taken into consideration that other factors, such as plant-derived inhibitors, may affect these findings.”          

L342-349: This paragraph is just a review of the literature. Please, add some discussion concerning your results.

The reviewer is correct. Thus the following paragraph was rewritten (Lines 344-347): “The excessive concentrations of sucrose or glucose in many cases reduce laccase production. However, in the cellulose-containing medium, the laccase was produced more efficiently [32]. Supporting that is our observation that the use of the minimal medium with the addition of 0.1% glucose resulted in relatively lower extracellular laccase activity (Figure 6).”
